# Resveratrol Alleviates the Early Challenges of Implant-Based Drug Delivery in a Human Glial Cell Model

**DOI:** 10.3390/ijms25042078

**Published:** 2024-02-08

**Authors:** Luise Schlotterose, François Cossais, Ralph Lucius, Kirsten Hattermann

**Affiliations:** Institute of Anatomy, Kiel University, 24118 Kiel, Germany; l.schlotterose@anat.uni-kiel.de (L.S.); rlucius@anat.uni-kiel.de (R.L.)

**Keywords:** brain implants, foreign body reaction, oxygen deprivation, glucose deprivation, reactive astrogliosis, neuroinflammation

## Abstract

Brain diseases are oftentimes life-threatening and difficult to treat. The local administration of drug substances using brain implants can increase on-site concentrations and decrease systemic side effects. However, the biocompatibility of potential brain implant materials needs to be evaluated carefully as implants can trigger foreign body reactions, particularly by increasing the microglia and astrocyte reactivity. To date, these tests have been frequently conducted in very simple in vitro models, in particular not respecting the key players in glial cell reactions and the challenges of surgical implantation characterized by the disruption of oxygen and nutrient supply. Thus, we established an in vitro model in which we treated human glial cell lines with reduced oxygen and glucose levels. The model displayed cytokine and reactive oxygen species release from reactive microglia and an increase in a marker of reactive astrocytes, galectin-3. Moreover, the treatment caused changes in the cell survival and triggered the production of hypoxia-inducible factor 1α. In this comprehensive platform, we demonstrated the protective effect of the natural polyphenol resveratrol as a model substance, which might be included in brain implants to ease the undesired glial cell response. Overall, a glial-cell-based in vitro model of the initial challenges of local brain disease treatment may prove useful for investigating new therapy options.

## 1. Introduction

Diseases of the brain, such as neurodegenerative diseases or tumors, are among the major causes of death globally [1]. Besides the challenges of timely diagnosis, one major issue in treating brain diseases is the low target concentrations of systemically administered drugs due to the privileged constitution of the brain. To overcome this challenge, drugs can be delivered in a site-specific manner using brain implants. The benefit of local administration is the chance to use drugs that are otherwise unable to pass through the blood–brain barrier, to reduce the systemic concentrations used, and, thereby, to minimize severe side effects [2,3].

To date, there are only a few examples of U.S. Food and Drug Administration (FDA)-approved brain implants for local drug delivery [4], as the implantation of foreign bodies into the brain tissue itself causes damage. The surgical procedure leads to bleeding and non-residual immune cell invasion and results in an inflammatory reaction [5]. Furthermore, the loss of perfusion leads to hypoxia and an impeded nutrient supply in the surrounding tissue. The long-term consequence is the formation of a glial scar encapsulating the implanted material and impeding drug release [6,7]. The foreign body response is mainly mediated by the microglia cells (resident immune cells [8]) and astrocytes (a heterogeneous group of glial cells with a variety of functions, e.g., structural support or brain homeostasis [9,10]). The microglia cells completely surround the implanted material within 24 h after implantation, limiting its connection to and exchange with the brain tissue [6]. Furthermore, they initiate further glial cell activation by releasing excitatory and proinflammatory factors [11,12]. Astrocytes are activated by reactive microglia signaling [13]. Once the astrocytes change in a reactive state, they produce high levels of extracellular matrix (ECM) proteins, which form the backbone of the glial scar [14,15]. In addition, they release saturated lipids, creating a neurotoxic micromilieu, leading to further neuronal death and axonal demyelination [16,17].

These tissue-damaging effects of the implantation process are—at least in part—comparable to the damage events in ischemic stroke. In particular, glial cell reactivity and rupture and dragging of the vasculature are observed in both settings. As a consequence of the latter, the surrounding tissue suffers from hypoxia and a lack of nutrient supply [18]. A commonly used in vitro model to study effects of ischemic strokes is oxygen/glucose deprivation (OGD) [19,20]. Given the parallels between ischemic stroke and brain implantation, OGD could also be used to elucidate the initial consequences caused by the implant insertion process in the brain and test substances to overcome these challenges.

In recent years, numerous substances have been analyzed in order to find therapeutics to ease neuroinflammation. Among them, resveratrol (5-[(*E*)-2-(4-hydroxyphenyl)ethenyl]benzene-1,3-diol) is a well-known polyphenol with antioxidant properties occurring in many plants, such as grapes, peanuts, and soy [21]. In a wide range of studies, resveratrol was found to milden inflammation, provide neuro- and cardio-protection, and have anti-cancer properties [22,23]. Additionally, resveratrol has been shown to have beneficial effects on the neurons in various stroke and OGD (and reperfusion) models [24,25,26,27]. These positive properties make resveratrol a potential candidate substance to be loaded onto brain implants in order to ease the unfavorable effects of implantation and glial scarring.

The aim of this study was to prove OGD as an easily applicable model to mimic the early challenges of implant-based therapies. It can be used to study their cellular and molecular consequences, including as a test platform to validate the beneficial effects of protective drug substances. Immortalized human astrocytes and microglia were challenged with OGD, and several common features of glial cell reactivity were monitored. Furthermore, resveratrol was used as a model substance to alleviate the OGD effects experimentally. This model will be useful to identify and overcome glial cell reactions when implemented in the testing and development of drug-loaded brain implants.

## 2. Results

### 2.1. Increased Levels of Reactive Oxygen Species in an Oxygen and Glucose Deprivation Model for Early Reactions after Implantation

Reactive oxygen species (ROS) production is one of the first responses to tissue injury and a main player in neuroinflammation [28]. To investigate whether our proposed in vitro model induces changes similar to the implantation process in brain tissue, the ROS production was analyzed in the first step at an early time point. Therefore, immortalized human microglia cells (HMC3) and immortalized human astrocytes (SVGA) were kept under oxygen/glucose deprivation conditions (OGD) as described in Section 4, and the ROS production was quantified using flow cytometry and using a ROS-sensitive fluorescent dye (Figure 1). As early as after 4 h, the OGD-treated samples showed an elevated ROS production compared to the controls, for both astrocytes (*p* = 0.001) and microglia (*p* = 0.0006). The cells were co-stimulated (or not) with 100 µM of resveratrol as a model substance to protect the glial cells, which is a commonly used concentration [29,30,31]. Co-stimulation with resveratrol reduced the ROS production almost to the control levels (normoxic and with glucose supply) for the astrocytes (Figure 1) and also significantly lowered the production levels in the microglia (*p* = 0.0181).

### 2.2. OGD Triggers Inflammatory Responses in the Microglia

To further test the OGD-induced microglial cell response, the mRNA and protein expression levels of the proinflammatory cytokines IL6 and IL1β were analyzed after 24 h of OGD exposure (Figure 2A,B). Cytokines play a key role in inflammatory processes and are released by the microglia after injuries [32]. Again, resveratrol (100 µM) was added to validate its effect on OGD-mediated inflammation. Quantitative polymerase chain reaction (qPCR) showed a significant increase in the mRNA production of both cytokines in the cells cultured under OGD conditions compared to the controls (IL6 *p* = 0.0251; IL1β *p* = 0.0003). These findings were confirmed at the protein level using an enzyme-linked immunosorbent assay (ELISA), where higher cytokine concentrations after OGD treatment in the cell supernatants for IL6 (*p* < 0.0001) and IL1β (*p* < 0.0001) were observed. In contrast, the OGD-triggered production of cytokines could be drastically reduced at the mRNA level (IL6 *p* = 0.0066; IL1β *p* = 0.0003) and protein level (IL6 *p* < 0.0001; IL1β *p* < 0.0001) in the samples supplemented with resveratrol.

### 2.3. Galectin-3 Upregulation Is Induced by OGD in the Astrocytes

Subsequently, the changes in the astrocytes after 24 h of OGD were investigated. Hence, qPCR (Figure 3A) and fluorescence immunocytochemistry (ICC) (Figure 3B,C) were conducted to evaluate the galectin-3 expression levels. Galectin-3 is a lectin that is involved in diverse cellular processes including cell–cell and cell–matrix interactions, as well as proliferation/apoptosis control. In the brain, the galectin-3 expression is increased in reactive astrocytes [33,34]. The results showed a significant increase in the mRNA levels of galectin-3 (*LGALS3*) in the cells incubated for 24 h under OGD conditions compared to the controls (*p* = 0.0047), and these effects were even more pronounced at the protein level, which was assessed via the quantification of the fluorescence signal intensity of the galectin-3 ICC (*p* = 0.0002). Treatment with 100 µM of resveratrol reduced the OGD-induced effects on the mRNA and protein expression, making it comparable to that in the control cells under the standard culture conditions (Figure 3A–C).

### 2.4. OGD Differentially Affects Glial Cell Growth and Apoptosis

To further validate our OGD model, we tested the effects on the apoptosis and growth rate of the microglia and astrocytes. Therefore, the activity of the key effector caspases 3 and 7 was analyzed after 24 h. The caspase 3/7 activity luminescence assay indicated OGD induced doubled caspase activity levels in the astrocytes (*p* = 0.0094) and equal activity levels in the microglia (Figure 4A). The addition of resveratrol (100 µM) reduced the basal levels of caspase 3/7 activity under normoxic conditions, and also mildend the OGD-induced caspase 3/7 activity, even below the basal levels. This effect was most pronounced in the microglia (control ± resveratrol: *p* < 0.0001; OGD ± resveratrol: *p* = 0.0026).

Despite the changes in the caspase 3/7 activity, the growth rate of the astrocytes after 24 h was hardly altered as a result of either OGD or resveratrol treatment (100 µM). Conversely, the microglia showed significantly decreased growth rates under OGD (*p* = 0.0151) that could in trend—but not statistically significantly—be alleviated by resveratrol treatment (Figure 4B).

### 2.5. HIF-1ɑ Accumulates upon OGD Treatment

The surgical procedure of implantation causes the disruption of perfusion and hypoxia in the surrounding tissue, inducing the transcriptional activity of hypoxia-inducible factor-1α (HIF-1α). HIF-1α signaling leads to changes in cell metabolism and angiogenesis [35,36]. As shown in Figure 5C, the ELISA confirmed an increase in the protein levels of HIF-1α in the astrocytes after 24 h of OGD treatment compared to the controls under the standard culture conditions (*p* = 0.004). In addition, quantification of the immunofluorescence intensity (Figure 5A,B) demonstrated that the HIF-1α expression in the OGD samples was significantly higher than in the controls (*p* = 0.0054). Moreover, an accumulation of HIF-1α in the nuclei was observed (Figure 5A image enlargements). Again, 100 µM of resveratrol could minimize the effects found in the astrocytes (ELISA: *p* = 0.002; ICC: *p* = 0.0098). On the other hand, there was no upregulation of HIF-1α after 24 h OGD observed for the microglia (Figure 5C). However, resveratrol (100 µM) treatment reduced both the OGD-induced (*p* = 0.0052) and basal (*p* = 0.023) levels of HIF-1α found in the microglia.

## 3. Discussion

In this study, we aimed to establish an elementary but comprehensive in vitro model for testing the glial–foreign body reactions when using potential brain implant components, mimicking the challenges of the early phase after the implantation process. Having survived the mechanical insult of the surgery, cells suffer from hypoxia and a reduced glucose supply due to the lack of perfusion, which leads to an inflammatory reaction and glial scarring [37]. We used stable and easy-to-culture glial cell lines of human origin and applied OGD, which is also used to investigate ischemia in vitro. In this study, OGD, applied to human astrocytes and microglia, was considered a platform for testing brain implant materials, drugs, and additives, implementing the effects of the initial implantation process. Indeed, induced release of ROS, higher levels of proinflammatory cytokines, astrocyte reactivity, HIF-1α accumulation, and changes in the glial cell viability were found under OGD conditions. Moreover, the antioxidant resveratrol was tested as a model substance to present the possibility of using the platform for drug discovery.

To investigate the response of the human glial cells to oxygen/glucose deprivation (OGD) the cells were kept in glucose-free medium, and the oxygen concentration was reduced to 3%. Since there is no specific defined oxygen concentration for hypoxia, a concentration of 3% was chosen to model a reduced but not completely impeded oxygen supply. This represents the circumstances after implantation, when several blood vessels are injured but others are still intact [38]. The control samples were kept under normoxic conditions, despite the oxygen concentration in the human brain being non-uniform and lower [39,40], to ensure optimal culturing conditions for the immortalized astrocyte and microglia cell lines.

The inflammatory responses to brain injuries often coincide with oxidative stress and with elevated levels of reactive oxygen species (ROS). The release of ROS leads to DNA damage, cell death, and neurotoxicity [41,42]. In this study, OGD was found to induce ROS production in the astrocytes and microglia at an early time point. While mainly the microglia are associated with ROS production [28], astrocytes are also known to contribute to ROS release in neuroinflammatory diseases, including Parkinson’s disease [43]. This is in contrast to a study by Kim et al. showing that astrocytes, after OGD treatment, emerged as controllers of microglia reactivity and secreted anti-inflammatory factors [44]. Nevertheless, resveratrol was able to reduce the ROS production in the astrocytes to levels almost equal to the control conditions. The ROS levels in the microglia were reduced but still higher than in the controls. These results support previous studies showing that resveratrol inhibits oxidative stress [45,46], indicating the contribution of astrocytes to ROS production but suggesting reactive microglia as the main source.

After an injury, the microglia produce not only ROS but also proinflammatory cytokines, which trigger neuroinflammation [47]. This study demonstrates the induction of mRNA expression and elevated protein levels of two main proinflammatory cytokines after 24 h of OGD treatment. Thus, OGD was found to promote an inflammatory response in the microglia. This correlates with prior studies where OGD was also found to trigger the production of inflammation-related factors in the microglia [48].

Astrocyte reactivity also contributes to the proinflammatory milieu [5] and is linked to reactive astrogliosis, causing specific functional and structural changes in the brain tissue. The lectin galectin-3 was used in this study as a marker for reactive astrocytes because of its modulating role in the Notch1 signaling pathway, which has a significant impact on astrocyte reactivity [34]. The slightly yet significantly increased mRNA levels of galectin-3 (*LGALS3*) in the OGD samples compared to the controls and the even more pronounced results of the ICC analysis emphasize the increased reactivity of the astrocytes in the presented OGD model. This is in line with previous results in primary rat astrocytes reported by Wang et al., showing glial scar-like characteristics after OGD treatment [49].

Resveratrol was able to return the reactive microglia and astrocytes back to a homeostatic state. These findings are in good agreement with recent studies showing the strong anti-inflammatory properties of resveratrol in animal models of neuroinflammation and stroke [50,51].

Glial cell reactivity is linked to changes in cell viability [12,52]. Differentiation between apoptosis and proliferation in the cells after 24 h of OGD treatment showed surprising results. The OGD treatment resulted in elevated caspase 3/7 activity in the astrocytes but not in the microglial cells. These observations indicate that microglia are more likely to survive an injury, while astrocytes initiate apoptosis in response to the same trigger. Therefore, microglia seem to be more robust, which is in agreement with microglia being the primary responders in defense after brain injury [53]. In contrast to the previous findings [12,37], OGD did not induce proliferation in the microglia. Instead, the microglia proliferated almost three times slower under OGD compared to the controls. These findings suggest that the microglial response to OGD seems to be variable, putatively depending on the cell line and experimental model [54].

Treatment with resveratrol showed a reduction in both the basal-level caspase 3/7 activity, which is observed under normoxic conditions with a normal glucose supply, and the OGD-increased caspase 3/7 activity in the glial cells. Resveratrol seems to provide a general protective effect on the microglia and astrocytes independent of exogenous challenges. Moreover, the observed effects of OGD on the cell growth in the microglia could partially be abolished by resveratrol.

The role of the HIF-1α pathway in oxygen regulation, energy metabolism, and survival have been investigated intensively. The accumulation of HIF-1α is known to be a direct outcome of hypoxia in cells [35,36]. In this study, OGD induced HIF-1α buildup in the astrocytes but not in the microglia. Previous studies [55] have shown an upregulation of HIF-1α also in the microglia. However, these studies used models with 0% oxygen; therefore, it seems that 3% oxygen, as used in this study, could still be sufficient for the microglia. This presents microglia to be less sensitive to lower oxygen concentrations than astrocytes. In relation to this result, Tadmouri et al. already found astrocytes to be activated more quickly, whereas the microglia required longer time spans of hypoxia to change to a reactive state [56]. Furthermore, resveratrol was able to inhibit the HIF-1α pathway, as previously shown in different studies focusing on cancer treatments [57,58].

Nevertheless, there are some limitations to this study. First, cell–cell interactions are of immense importance when it comes to pathological changes. Individual cultures of astrocytes or microglia lack these interactions. Secondly, despite the use of human cell lines rather than rodent cell lines, immortalized cell lines come with functional and biological issues. The use of, for example, induced pluripotent stem cells (iPSCs) could be a step toward an even more comprehensive model. Thus, the presented model using a glial co-culture driven by iPSCs would also be worth exploring but connected to different ethical considerations and much higher costs and efforts. 

To summarize, the suggested human glial cell model can serve as an advanced in vitro platform for testing potential brain implant materials, loading drugs, and additives like coatings. Furthermore, the platform is easily available to many laboratories and implements several aspects of the implantation process like elevated ROS, HIF-1α, and cytokine levels; glial cell reactivity; and the affection of cell proliferation and survival. Further read-outs can be added according to the actual testing setup. The effects of resveratrol, which was shown to be protective of the neurons in ischemic animal models as well as neuronal OGD culture models [25,26,59,60], could be successfully validated in this glial-cell-based OGD model.

## 4. Materials and Methods

### 4.1. Cell Culture

The human microglia cell line HMC3 (Cat. CRL-3304, RRID: CVCL_II76) was purchased from the American Type Culture Collection (ATCC, Manassas, VA, USA). The human fetal astrocyte cell line SVGA was kindly provided by the group of Christine Hanssen Rinaldo, the University Hospital of North Norway [61] with the permission of W.J. Altwood [62]. The cells were grown in Dulbecco’s modified Eagle’s medium (Cat. 41965, DMEM, 25 mM glucose, Thermo Fisher Scientific, Darmstadt, Germany) supplemented with 10% fetal bovine serum (PAN-Biotech GmbH, Aidenbach, Germany), 1% penicillin-streptomycin (10,000 U/mL; Thermo Fisher Scientific), and 2 mM additional L-glutamine (Carl Roth, Karlsruhe, Germany) and incubated at 5% CO_2_/37 °C. The cells were routinely checked for mycoplasma contamination using a mycoplasma-specific PCR (Cat. 11-1100, Venor^®^GeM Classic; Minerva Biolabs^®^, Berlin, Germany).

### 4.2. Oxygen/Glucose Deprivation Model

The human microglia (HMC3) and astrocytes (SVGA) were seeded two days prior to stimulation to adhere at 5% CO_2_/37 °C. On the day of the experiment, the cells were washed once with phosphate-buffered saline (PBS). The medium was renewed for the controls, and for the treatment group, changed to 1 mL of OGD medium, a DMEM-based glucose-free medium (Cat. A14430, DMEM no-glucose, Thermo Fisher Scientific), with 1% penicillin–streptomycin (10,000 U/mL; Thermo Fisher Scientific) and 3.97 mM L-glutamine (Carl Roth), supplemented or unsupplemented with 100 µM of resveratrol (Cat. R5010, Sigma-Aldrich/Merck, Taufkirchen, Germany) dissolved in polyethylene glycol 400 (PEG400, Caesar & Loretz GmbH, Hilden, Germany). The PEG400 did not have a significant effect on the cells used in this study (Appendix A, Supporting Information). The cells were kept at 3% O_2_/5% CO_2_/37 °C for the times indicated for each experiment in an incubator (Memmert ICO150 with AtmoCONTROL, Schwabach, Germany). Afterward, the media and cells were collected for subsequent analysis.

### 4.3. Cell Apoptosis and Proliferation Assay

For the cell apoptosis assay, the astrocytes (SVGA) and microglia (HMC3) were cultured on 96-well plates (7500 cells/100 µL medium/well). The cells were treated under OGD conditions for 24 h, and subsequently apoptosis was analyzed using the Caspase-Glo^®^ 3/7 Assay (Cat. G8090, Promega, Madison, WI, USA) according to the manufacturer’s instructions. The plates were read using a Tecan GENios microplate reader (Tecan Group Ltd., Männedorf, Switzerland). Proliferation was determined by counting the cells seeded into 6-well plates (80,000 cells/well) using the TC20 Automated Cell Counter (Bio-Rad, Feldkirchen, Germany) after the OGD treatment. Proliferation was calculated as an n-fold amount of the initially seeded cell number.

### 4.4. Enzyme-Linked Immunosorbent Assay (ELISA)

For quantification of the interleukin-1beta (IL1β) and interleukin-6 (IL6) levels in the cell supernatant, the microglia (HMC3) were seeded into 6-well plates (80,000 cells/1 mL medium/well). Following 24 h of OGD treatment, the cytokine levels were measured in technical duplicates using the human interleukin-1beta ELISA (Cat. 31670019) and human interleukin-6 ELISA (Cat. 31670069, both ImmunoTools, Friesoythe, Germany) according to the manufacturer’s instructions.

For quantification of the hypoxia-inducible factor-1α (HIF-1α) levels, the astrocytes (SVGA) and microglia (HMC3) were seeded into T25 tissue culture flasks (300,000 cells/4 mL medium/flask). After 24 h of OGD treatment, the cells were lysed in modified RIPA buffer (50 mM Tris, 100 mM NaCl, 5 mM EDTA, 1% NP-40, 1 mM Na_3_VO_4_, 0.8 mM CoCl_2_, 1x cOmplete™ Mini Protease Inhibitor Cocktail (Roche, Sigma-Aldrich/Merck, Taufkirchen, Germany)). The total protein concentration was determined using the Pierce™ BCA Protein Assay Kit (Cat. 23225, Thermo Fisher Scientific), and HIF-1α was quantified using the HIF1A Human ELISA Kit (Cat. EHIF1A2, Thermo Fisher Scientific) in technical duplicates according to the manufacturer’s instructions. The plates were read using the BioTek Epoch microplate reader (Agilent Technologies, Santa Clara, CA, USA).

### 4.5. Reactive Oxygen Species

For quantification of the reactive oxygen species (ROS) levels, the astrocytes (SVGA) and microglia (HMC3) were cultured on 12-well plates (40,000 cells/500 µL medium/well). Before the OGD treatment, the cells were stained using the Total Reactive Oxygen Species (ROS) Assay Kit 520 nm (Cat. 88-5930-74, Thermo Fisher Scientific) according to the manufacturer’s instructions. After 4 h of treatment, the cells were fixed for 20 min in 4% paraformaldehyde (PFA) in PBS at room temperature and washed with PBS, and the dye fluorescence was analyzed using FACSCanto II (Biosciences, Heidelberg, Germany) with 488 nm excitation and detection with a 530/30 band-pass filter. The results were analyzed and plotted using FlowJo™ software v10.7.1 (BD Life Sciences, Heidelberg, Germany).

### 4.6. Quantitative PCR

The astrocytes (SVGA) and microglia (HMC3) were seeded into 6-well plates (80,000 cells/well). After 24 h of OGD treatment, the cells were harvested and homogenized using TRI Reagent^®^ (Cat. T9424, Sigma-Aldrich/Merck), and total RNA was isolated following the manufacturer’s protocol. The genomic DNA was digested using RNase-free DNase (1 U/µL, Thermo Fisher Scientific), and the cDNA was synthesized using Thermo Scientific’s RevertAid RT Kit (Cat. K1691, Thermo Fisher Scientific). TaqMan primers and probes (Thermo Fisher Scientific) and the HOT FIREPol^®^ Probe Universal qPCR Mix (Cat. 08-17-00001, Solis BioDyne, Tartu, Estonia) were utilized to analyze the samples using the ABI PRISM 7500 sequence detection system (Applied Biosystems, Waltham, MA, USA). The analyzed genes were *RNA18S* (Hs99999901_s1), *IL6* (Hs00985639_m1), *IL1β* (Hs01555410_m1), and *LGALS3* (Hs00174774_m1).

The cycle threshold values (CT) were determined, and the ∆CT values = CT [gene of interest] − CT [18S rRNA] were calculated. Due to the logarithmic reaction mode, a ∆CT value of 3.33 corresponds to a one order of magnitude lower gene expression compared to 18S rRNA. For OGD-induced mRNA regulation, the ∆∆CT values were calculated as follows: ∆∆CT = 2 ˄–(∆CT[stimulus] − ∆CT[control]).

### 4.7. Immunocytochemistry

The astrocytes (SVGA) and microglia (HMC3) were seeded onto glass coverslips (80,000 cells/coverslip). Following 24 h of OGD treatment, the cells were briefly rinsed with PBS and fixed for 10 min in 4% PFA at room temperature. Immunocytochemistry was carried out after permeabilization with 0.1% Triton X-100 (Sigma-Aldrich/Merck) in PBS for 5 min at room temperature and blocking for 60 min in BSA (0.5%, Biomol, Hamburg, Germany) and glycine (0.5%, Carl Roth) in PBS. Incubation with the primary antibodies was performed in PBS at 4 °C overnight. The following primary antibodies were used: anti-galectin-3 (goat, 1:100; Cat. AF1154, Bio-Techne GmbH, Wiesbaden-Nordenstadt, Germany) and anti-HIF-1α (mouse, 1:100; Cat. NB100-131, Novus Biologicals, Wiesbaden, Germany). Afterward, the cells were incubated with Alexa Fluor 488- or 555-labeled secondary antibodies against the respective host species (donkey IgG, 1:1000, Thermo Fisher Scientific) at 37 °C for 1 h, and their nuclei were counterstained with DAPI (Sigma-Aldrich/Merck). The slides were embedded with Shandon Immu-Mount™ (Cat. FIS9990402, Thermo Fisher Scientific). For the secondary antibody controls, the primary antibodies were omitted (Appendix A). Imaging was carried out using the Keyence BZ-800 fluorescence microscope (Keyence GmbH, Neu-Isenburg, Germany). The fluorescence intensity of two areas from each experiment was quantified using ImageJ v1.49 (RRID: SCR_003070) [63].

### 4.8. Statistical Analysis

Statistical analysis was performed using GraphPad Prism v9.4.1 (RRID: SCR_002798). All the results are presented as mean values ± standard deviation (SD), and the numbers of biological replicates are stated in the respective figure legends. No blinding or testing for outliers was performed. Statistically significant differences were evaluated using one-way analysis of variance (ANOVA), followed by Tukey’s post hoc test for comparisons between multiple groups, and probability values *p* < 0.05 were considered statistically significant.

## 5. Conclusions

In conclusion, we established this in vitro model using human astrocytes and microglia cells to mimic the initial glial cell responses to the implantation process. This model includes the early challenges faced immediately after the surgical procedure, and it is simpler, more cost-effective, and more ethically justifiable compared to animal studies. This allows for a higher throughput, e.g., in screening implant properties and drug substances that could counteract the harmful effects following implantation, and could thus pave the way for the development of site-specific drug delivery systems. Furthermore, the model will be useful for understanding the mechanisms involved in inflammatory processes within the brain.

## Figures and Tables

**Figure 1 ijms-25-02078-f001:**
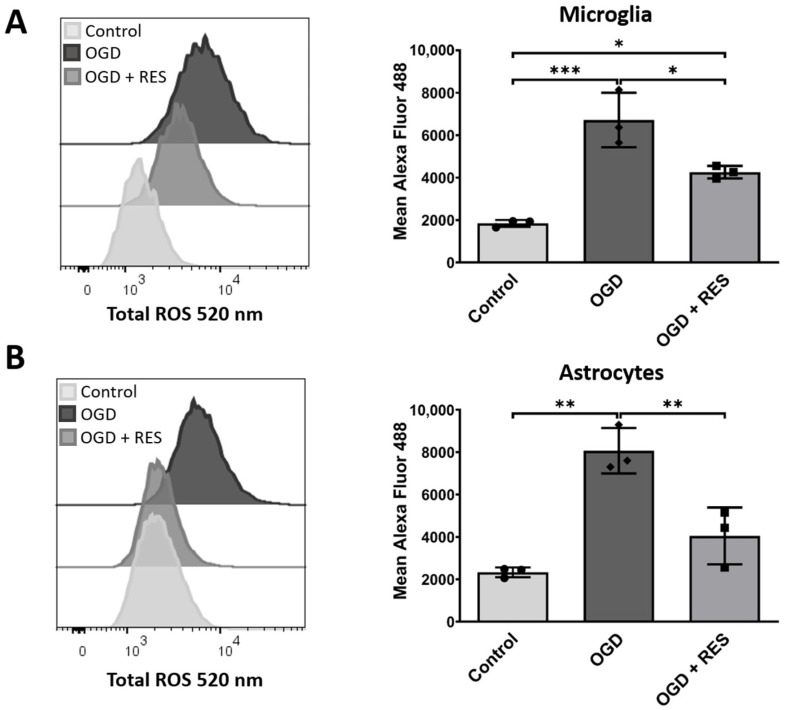
Production of reactive oxygen species increases during OGD. (**A**,**B**) Representative histograms of staining for total ROS in unstimulated, 24 h OGD-activated, and OGD-activated and resveratrol-treated glial cells. Mean fluorescence intensity measured using flow cytometry reveals higher intensity for samples after 4 h OGD treatment in ROS-labeled (**A**) microglia and (**B**) astrocytes. Cells were co-treated or untreated with 100 µM resveratrol (RES) and compared to untreated control. *n* = 3, number of independent cell cultures, indicated by separate dots (normoxic, without resveratrol), rhombs (OGD without resveratrol) and squares (OGD plus resveratrol); * *p* < 0.05, ** *p* < 0.01, and *** *p* < 0.001.

**Figure 2 ijms-25-02078-f002:**
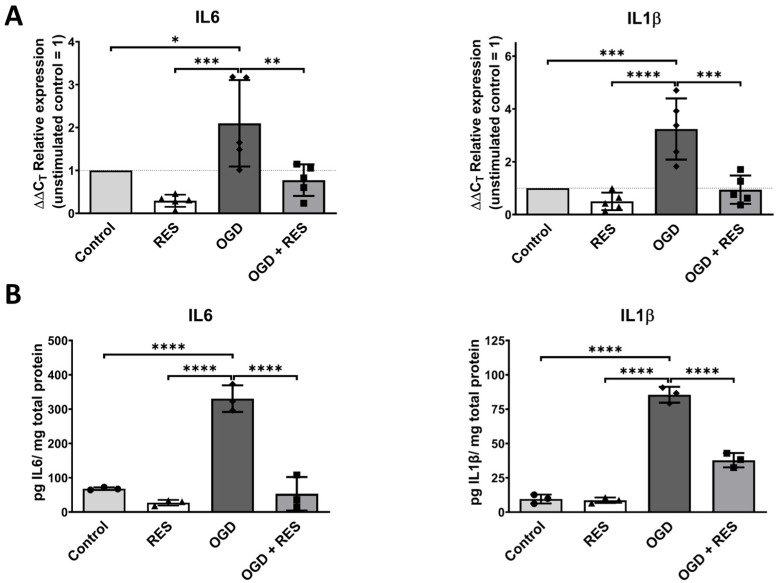
Proinflammatory effects of OGD on human microglia. (**A**) qPCR analysis showing that 24 h OGD treatment increases expression of proinflammatory cytokines IL6 and IL1β. (**B**) ELISA confirms higher levels of cytokines IL6 and IL1β after 24 h of OGD treatment found in supernatant, normalized to total protein content. Microglia were co-treated or untreated with 100 µM resveratrol (RES) and compared to untreated control. (**A**) *n* = 5, number of independent cell cultures, indicated by separate triangles (normoxic plus resveratrol), rhombs (OGD without resveratrol) and squares (OGD plus resveratrol), the dotted lines indicates the normalization level of controls (normoxic without resveratrol); (**B**) *n* = 3, number of independent cell cultures, indicated by separate dots (normoxic without resveratrol), triangles (normoxic plus resveratrol), rhombs (OGD without resveratrol) and squares (OGD plus resveratrol); * *p* < 0.05, ** *p* < 0.01, *** *p* < 0.001 and **** *p* < 0.0001.

**Figure 3 ijms-25-02078-f003:**
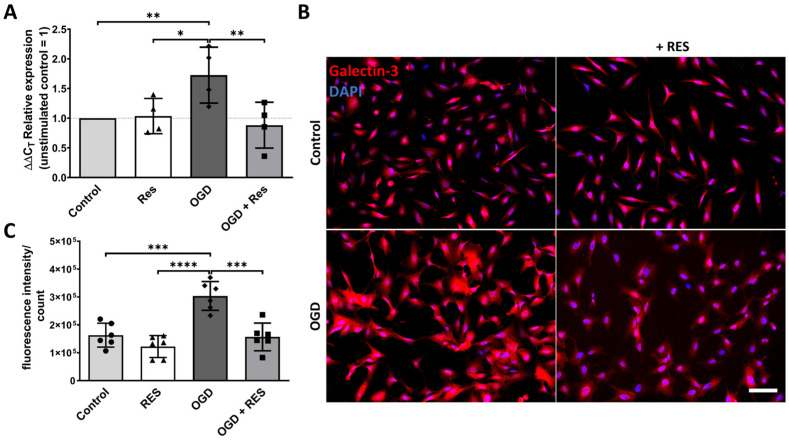
Increased reactivity of human astrocytes after OGD. (**A**) qPCR analysis showing that 24 h OGD treatment increases expression of astrocyte reactivity marker galectin-3. (**B**) Representative immunofluorescence staining images of galectin-3 (red) and nuclei (DAPI, blue) support findings at protein level (scale bar: 50 µm). (**C**) Corresponding quantification of fluorescence intensity normalized to cell numbers. Astrocytes were co-treated or untreated with 100 µM resveratrol (RES) and compared to untreated control. (**A**) *n* = 4, number of independent cell cultures, indicated by separate triangles (normoxic plus resveratrol), rhombs (OGD without resveratrol) and squares (OGD plus resveratrol), the dotted lines indicates the normalization level of controls (normoxic without resveratrol); (**B**,**C**) *n* = 3, number of independent cell cultures, indicated in (C) by separate dots (normoxic without resveratrol), triangles (normoxic plus resveratrol), rhombs (OGD without resveratrol) and squares (OGD plus resveratrol); * *p* < 0.05, ** *p* < 0.01, *** *p* < 0.001 and **** *p* < 0.0001.

**Figure 4 ijms-25-02078-f004:**
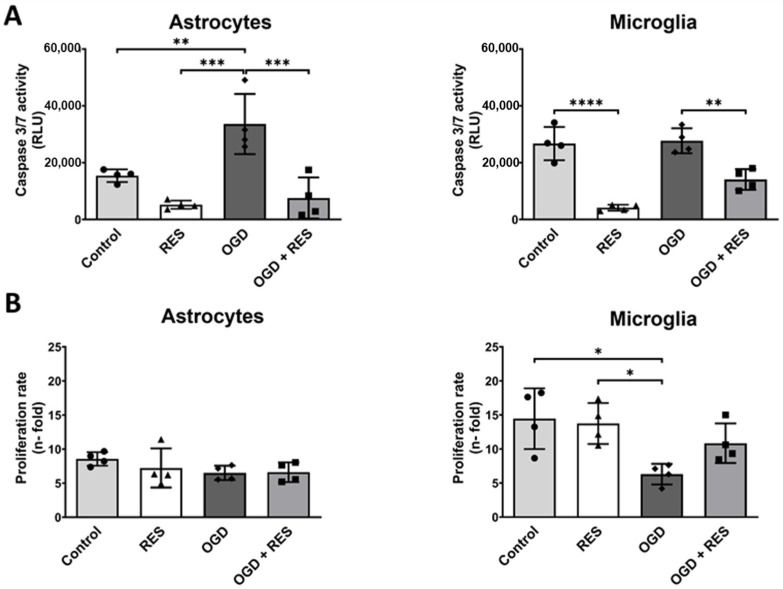
OGD’s effects on survival and proliferation of human microglia and astrocytes. (**A**) Effects on caspase 3/7 activity, as a sign of apoptosis, show only an effect on astrocytes after 24 h OGD (RLU = relative light units). Resveratrol, however, reduced caspase 3/7 activity in both controls and OGD cultures. (**B**) n-fold changes in cell proliferation after 24 h OGD treatment reveal decelerated growth of microglia cells. Cells were co-treated or untreated with 100 µM resveratrol (RES) and compared to untreated control. (**A**,**B**) *n* = 4, number of independent cell cultures, indicated by separate dots (normoxic without resveratrol), triangles (normoxic plus resveratrol), rhombs (OGD without resveratrol) and squares (OGD plus resveratrol); * *p* < 0.05, ** *p* < 0.01, *** *p* < 0.001 and **** *p* < 0.0001.

**Figure 5 ijms-25-02078-f005:**
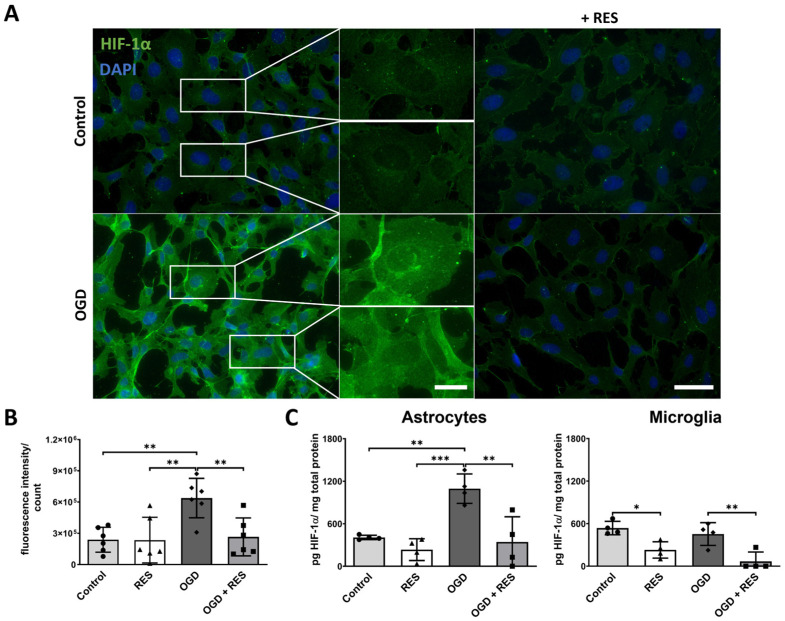
HIF-1ɑ accumulation upon OGD in human astrocytes. (**A**) Representative immunofluorescence staining images of HIF-1ɑ (green) and nuclei (DAPI, blue) in astrocytes (scale bar: 50 µm) showed accumulation of HIF-1ɑ and enhanced HIF-1ɑ staining in nuclei (scale bar: 10 µm). (**B**) Corresponding quantification of fluorescence intensity normalized to cell number. (**C**) Quantification of HIF-1ɑ protein in astrocytes and microglia cell lysates using ELISA confirms higher levels of HIF-1ɑ in astrocytes. Cells were co-treated or u with 100 µM resveratrol (RES) and compared to untreated control. (**A**,**B**) *n* = 3, number of independent cell cultures; (**C**) *n* = 4, number of independent cell cultures; (**B**,**C**) separate dots (normoxic without resveratrol), triangles (normoxic plus resveratrol), rhombs (OGD without resveratrol) and squares (OGD plus resveratrol) indicate independent experiments; * *p* < 0.05, ** *p* < 0.01, and *** *p* < 0.001.

## Data Availability

All collected data is shown as separate data points for any individual experiment. The original measurement data that support the findings of this study are available from the corresponding author upon request. These original data are not publicly available as they do not comprise further information.

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
