# Peer review of "Resveratrol Alleviates the Early Challenges of Implant-Based Drug Delivery in a Human Glial Cell Model"

_ijms, 2024, doi:10.3390/ijms25042078_

Round 1
Reviewer 1 Report
Comments and Suggestions for Authors
The Manuscript presents an in vitro glia cell model with oxygen/glucose deprivation conditions (OGD), which may be useful for testing of potential brain implant materials, loading drugs and substances that reduce inflammation. Differential responses of astrocytes and microglia to OGD and the protective effect of the flavonoid resveratrol were convincingly demonstrated. For example, OGD is not toxic to microglia, does not lead to apoptosis, but inhibits proliferation. Whereas it is toxic for astrocytes, causes apoptosis and does not affect the proliferation rate. The study was carried out at a good experimental level. The data obtained are of both theoretical and practical interest. However, there are a number of comments.
1. It is necessary to indicate where the cells were incubated at 3% oxygen (type of hypoxic chamber or hypoxic incubator).
2. The authors are absolutely right that in OGD simulations, 3% oxygen is more physiological than 0%. In brain tissue, the partial pressure of oxygen is significantly lower than 20%, for example in the cortex it is about 5%. It is probably necessary to somehow highlight this fact and justify the choice of control conditions. In the future, it would be interesting to compare the effects at different oxygen partial pressures, including 5%, 3% and 1%.
3. Line 146-148 “Conversely, microglia showed significantly decreased growth rates under OGD (p = 0.0151) that, however, could partly be reversed by resveratrol treatment (Fig. 4 B).“ Is the reversible effect of resveratrol reliable in this case? There is no asterisk on the graph.
4. Why does the caption to Figure 1 indicate 24 hours of incubation, but the method of ROS level detection only says 4 hours of incubation?
5. The authors give the change in the number of cells over 24 hours as the rate of proliferation. However, some cells may die because OGD induces apoptosis in astrocytes. It would be interesting to evaluate how OGD and resveratrol affect the rate of division of living cells.
Comments on the Quality of English LanguageMinor language corrections are necessary to clarify some wording.
Reviewer 2 Report
Comments and Suggestions for Authors
The paper examines the use of an in vitro model to study glial cells in implants that could be used for drug delivery to the CNS and the effect of resveratrol on the injury responses to hypoxia and hypoglycemia. The paper is interesting and the figures are well organized. The main concern is the arrangement of the presentation should be adjusted to improve grammar and organizational flow.
Specifically:
The paper goes back and forth regarding the model, the injury stimulus and the effect of the resveratrol intervention. Starting with the introduction this order should be maintained including the last line of the introduction.
The discussion has numerous sentences that repeat the results in paragraph 2, 3, 4. There should be more discussion regarding implications of the results and how they conform or not to prior studies etc.
There are several instances where sentence structure is not clear such as last sentence of abstract, lines 51/52, line 58, and last paragraph of introduction.
Comments on the Quality of English Language
See above suggestions
